# Overexpression of a Voltage-Dependent Anion-Selective Channel (VDAC) Protein-Encoding Gene, *MsVDAC*, from *Medicago sativa* Confers Cold and Drought Tolerance to Transgenic Tobacco

**DOI:** 10.3390/genes12111706

**Published:** 2021-10-27

**Authors:** Mei Yang, Xinhang Duan, Zhaoyu Wang, Hang Yin, Junrui Zang, Kai Zhu, Yumeng Wang, Pan Zhang

**Affiliations:** Department of Grassland Science, College of Animal Science and Technology, Northeast Agricultural University, Harbin 150030, China; yangmei199611@163.com (M.Y.); duanxinhang96@163.com (X.D.); wzy591128724@163.com (Z.W.); yinhang361@163.com (H.Y.); zangjunrui@126.com (J.Z.); zhukai09030018@163.com (K.Z.); rubywangl999@163.com (Y.W.)

**Keywords:** alfalfa, voltage-dependent anion channels, ROS scavenging, osmotic homeostasis, stress-responsive genes

## Abstract

Voltage-dependent anion channels (VDACs) are highly conserved proteins that are involved in the translocation of tRNA and play a key role in modulating plant senescence and multiple pathways. However, the functions of *VDACs* in plants are still poorly understood. Here, a novel *VDAC* gene was isolated and identified from alfalfa (*Medicago sativa* L.). *MsVDAC* localized to the mitochondria, and its expression was highest in alfalfa roots and was induced in response to cold, drought and salt treatment. Overexpression of *MsVDAC* in tobacco significantly increased MDA, GSH, soluble sugars, soluble protein and proline contents under cold and drought stress. However, the activities of SOD and POD decreased in transgenic tobacco under cold stress, while the O_2_
^-^ content increased. Stress-responsive genes including *LTP1*, *ERD10B* and *Hxk3* were upregulated in the transgenic plants under cold and drought stress. However, *GAPC*, *CBL1*, *BI-1*, *Cu/ZnSOD* and *MnSOD* were upregulated only in the transgenic tobacco plants under cold stress, and *GAPC*, *CBL1*, and *BI-1* were downregulated under drought stress. These results suggest that *MsVDAC* provides cold tolerance by regulating ROS scavenging, osmotic homeostasis and stress-responsive gene expression in plants, but the improved drought tolerance via *MsVDAC* may be mainly due to osmotic homeostasis and stress-responsive genes.

## 1. Introduction

Plants are sessile organisms, and in nature, they have to cope with abiotic stresses and environmental pressures to maintain their growth [1]. Drought stress is the main environmental stress in China; drought stress dramatically limits plant growth and development and significant decreases crop yields and quality [2]. In addition, cold stress is another major limiting environmental factor that influences plant growth, development, productivity and geographical distribution. In plants, cold stress can cause physiological, metabolic and molecular changes that lead to membrane rearrangement, increased osmotic pressure, accumulation of reactive oxygen species (ROS), and abnormal functions of mitochondria and other organelles [3,4]. Consequently, plants have developed a complex tolerance mechanism to minimize the negative effects of drought and cold stress, including changes at the molecular, physiological, and whole-plant levels to the ecosystem level [5,6]. Identifying the proteins or genes controlling these changes may lead to rapid genetic improvement for drought and cold tolerance of crops.

Voltage-dependent anion channels (VDACs) are highly conserved proteins that promote and regulate the flow of metabolites and mediate multiple pathways in both animals and plants [7]. In animals, *VDACs* have been extensively studied for their participation in the response mechanism involving ROS. In plants, *VDACs* are major components of pathways involved in the translocation of transfer RNA (tRNA) through the mitochondrial outer membrane. Cell death that occurred in the tobacco cell line BY2 and in the leaves of tobacco plants heterologously expressing a rice *VDAC* demonstrated that the ratio of *VDAC* and hexokinase expression and the interaction of these proteins play a key role in modulating senescence-related pathways in plants [8]. *Arabidopsis VDACs* have been shown to participate in the regulation of respiration, the balance of ROS, and the stress tolerance of yeast [9]. *AtVDAC1* is not only necessary for normal growth but also important for disease tolerance via its ability to regulate the generation of hydrogen peroxide (H_2_O_2_). *AtVDAC2* and *AtVDAC4* are important for various physiological functions, including leaf development, the steady state of the mitochondrial membrane potential, and pollen development [10]. In *Brassica rapa*, the expression of *BrVDAC* showed a strong increase under drought, cold and salt stress and was highest in the leaves of young seedlings [11]. A study on the subcellular distribution of a plant VDAC-like protein between plastids and mitochondria in green and nongreen tissue demonstrated that this VDAC isoform targets mitochondria exclusively and not plastids [12].

Alfalfa (*Medicago sativa* L.) is an important cultivated forage legume crop species worldwide and plays a major role in agricultural sustainability [13]. It has the characteristics of drought resistance, cold tolerance, and high protein content, which makes it the basis for the development of herbivorous animal husbandry [14,15]. With the development of global dairy production, the demand for alfalfa is likely to continue to grow [16]. Although the deep root system of this species can avoid drought-induced damage on semidry land, the climate in northern China, particularly the low temperatures during winter and the low amount of rainfall, are the main factors affecting the growth of alfalfa. Therefore, improving drought and cold tolerance are key goals of alfalfa breeders in the context of global climate change.

To date, large-scale studies intended to elucidate drought and cold tolerance mechanisms in alfalfa have been performed via physiological, genomic and agronomic approaches [17]. Many genes have been screened, but their functions are still poorly understood, especially those of *VDACs*, which have rarely been studied in plants, especially alfalfa. Thus, the aim of this study was to investigate the role of the *MsVDAC* gene in drought and cold tolerance and to provide evidence for its potential uses in improving drought and cold tolerance in forage legumes. The expression of *MsVDAC* in response to abiotic stress and transgenic plants overexpressing *MsVDAC* under drought and cold resistance was analysed. 

## 2. Materials and Methods

### 2.1. Plant Growth and Experimental Treatments

Seeds of alfalfa (*M. sativa* L. cv. Dongnong No. 1) were disinfected with a 5% sodium hypochlorite solution (NaClO) for 5 min and a 70% ethyl alcohol solution for 30 s and then rinsed in distilled water 4–5 times. The seeds were then germinated on moistened filter paper in culture dishes in a growth chamber at 25 °C and 70% humidity in the dark. Five-day-old seedlings were transplanted into plastic pots filled with vermiculite in a greenhouse. The greenhouse temperature was 25 °C and 20 °C, and the relative humidity was 55% and 70% during the day and night, respectively. All the seedlings were watered with 1/2 strength Hoagland nutrient solution every two days. The roots, leaves, and racemes of 6-week-old seedlings were sampled and stored at −80 °C for tissue-specific expression analysis of *MsVDAC*. Then, the seedlings were divided into three groups. The first group was placed in a 4 °C growth chamber for cold treatment. The second group was watered with a nutrient solution consisting of 20% PEG6000 for drought treatment. The third group was watered with nutrient solution containing 150 mM NaCl for salt treatment. All the roots and leaves were harvested at 0 (control), 3, 6, 12, 24 and 48 h after treatment and subsequently stored at −80 °C to evaluate the expression of *MsVDAC* in response to abiotic stresses.

### 2.2. Identification and Conserved Motif Analysis of VDAC Genes in Alfalfa

To identify the *VDAC* genes in alfalfa, the amino acid sequence of all *VDACs* in *Arabidopsis* were downloaded from the *Arabidopsis* Information Resource (TAIR, https://www.arabidopsis.org/, accessed on 23 October 2021). Then, the sequence information was used to blast homologs with the genome of *M. sativa* L. cv. Zhongmu No.1 (https://figshare.com/articles/dataset/Medicago_sativa_genome_and_annotation_files/12623960, accessed on 23 October 2021). The sequences with E-values greater than 1 × 10^−6^ or without VDAC domain predicted with SWISS-MODEL (https://swissmodel.expasy.org/, accessed on 23 October 2021) were removed. The phylogenetic tree was constructed based on the predicted sequence via MEGA software (version 7.0). The conserved motifs were analysed by the online program SMART (http://smart.embl.de/smart/set_mode.cgi?GENOMIC=1, accessed on 23 October 2021).

### 2.3. Isolation and Cloning of MsVDAC

Total RNA was extracted from leaves with an Ultrapure RNA Kit (CWBIO, Beijing, China) according to the manufacturer’s instructions. The integrity of total RNA was confirmed via electrophoresis on a 1% agarose gel, and the concentration and integrity were assessed by NanoPhotometer (Implen, Munich, Germany). Then, the extracted RNA was reverse-transcribed into cDNA using a HiScript II 1st Strand cDNA Synthesis Kit (+gDNA wiper) (Vazyme, Nanjing, China) according to the manufacturer’s instructions. The coding DNA sequence (CDS) of *MsVDAC* was obtained via PCR with degenerate primers (forward primer, 5′-ATGGCTAACGGTCCAGCAC-3′; reverse primer, 5′-TTAAGGCTTGAGAGAAAGAGCG-3′) designed based on the *VDAC* sequence of *M. truncatula* retrieved from the National Center for Biotechnology Information (NCBI) and via Primer Express 5.0 software (Applied Biosystems, Carlsbad, CA, USA) using a 2 × Unique^TM^ Taq Master Mix (with Dye) (Novogene, Beijing, China) following the manufacturer’s instructions. The PCR product was subsequently cloned and inserted into a PCE2 TA/Blunt-Zero-T vector (Vazyme, Nanjing, China) for sequencing.

### 2.4. Sequence Analysis of MsVDAC

VDAC homologues were identified via the EMBL-EBI (https://www.ebi.ac.uk/, accessed on 27 January 2021) BLAST program. The open reading frames (ORFs) were predicted with NCBI ORF finder. Multiple sequence alignment was performed using DNAMAN software (version 7.0). A phylogenetic tree was constructed by MEGA software (version 7.0) with the neighbour-joining (NJ) method. The reliability of each tree was determined by bootstrap analysis with 1000 replicates. The secondary structure of the *MsVDAC* protein was predicted by the online software SOPMA (http://pbil.ibcp.fr, accessed on 27 January 2021), and the 3D structure of the protein was predicted by SWISS-MODEL (https://swissmodel.expasy.org/, accessed on 27 January 2021). The predicted subcellular localization of the *MsVDAC* protein was analysed by the online program CELLO 2.5 (http://cello.life.nctu.edu.tw/, accessed on 27 January 2021). Similarly, the conserved motifs were analysed by the online program SMART (http://smart.embl.de/smart/set_mode.cgi?GENOMIC=1, accessed on 27 January 2021), and the physicochemical properties were analysed by the online program ProtParam.

### 2.5. Subcellular Localization Assays

*MsVDAC* was amplified using specific primers (5′-cgggggactcttgacgagctc ATGGCTAACGGTCCAGCAC-3′; the SacI restriction site is underlined) and (5′-catgtcgactctagaggatcc AGGCTTGAGAGAAAGAGCGAG-3′; the BamHI restriction site is underlined). The amplicons were double-digested with SacI and BamHI and then ligated to a pCAMBIA1300-GFP vector. The ligated products were transformed into *Escherichia coli* strain *DH5α* competent cells. The plasmids were subsequently extracted from positive clones using a FastPure Plasmid Mini Kit (Vazyme, Nanjing, China) and sequenced. A 35S-MsVDAC-GFP positive recombinant plasmid was introduced into *Agrobacterium tumefaciens* strain EHA105. Then, the solution was injected into the blade back of *Nicotiana benthamiana* leaves; after 48 h of culture in the dark, the leaves were observed via a laser scanning confocal microscope.

### 2.6. Quantitative Real-Time PCR (qPCR) Analysis

The tissue-specific expression of *MsVDAC* and its expression in response to abiotic stress were investigated using qPCR assays. Total RNA was isolated from each harvested tissue with an Ultrapure RNA Kit (CWBIO, Beijing, China) according to the manufacture’s instructions. The integrity of total RNA was confirmed via electrophoresis on a 1% agarose gel, and the concentration and integrity were assessed by NanoPhotometer (Implen, Munich, Germany). Subsequently, cDNA was reverse transcribed using the HiScript® II Q RT SuperMix for qPCR (+gDNA wiper) Kit (Vazyme, Nanjing, China) based on the manufacturer’s instructions. Next, 2 µL RNA (1 µg) and 4 µL 4 × gDNA wiper Mix were mixed with 10 µL RNA-free water, and incubated at 42 °C for 2 min to remove genomic DNA. Then, the reaction solution was mixed with 4 µL 5 × HiScriptII qRT SuperMix II, and the reverse transcription reaction was conducted at 50 °C for 15 min and 85 °C for 5 s.

The qPCR primers of *MsVDAC* (forward primer, 5′-TAATGCCGGGATTGCCTTCA-3′; reverse primer, 5′-CTGGGCGGTCCACATAATGAC-3′) were designed based on the CDS of *MsVDAC* and target-length of qPCR product was 107 bp. The alfalfa *β-actin* (*MsActin*) gene (Appendix A) was used as an internal reference control [18]. qRT-PCR was conducted via Quantagene q225 Real Time PCR apparatus (Novogene, Beijing, China) using the ChamQTM Universal SYBR® qPCR Master Mix (Vazyme, Nanjing, China) according to the manufacturers’ instructions. Then, 10 µL 2 × Cham Q Universal SYBR qPCR Master Mix, 1 µL Primer F, 1 µL Primer R, 1 µL cDNA and 7 µL dd H_2_O were mixed together. The reaction procedure of qPCR was: pre-denaturation at 95 °C for 30 s, 40 cycles of cyclic-reaction at 95 °C for 10 s and 58 °C for 30 s, followed by dissociation curve at 95 °C for 15 s, 60 °C for 60 s and 95 °C for 15 s. Three independent biological replicates and three replicate reactions were included for each sample in this experiment. The relative expression levels of *MsVDAC* were calculated using the comparative cycle threshold (Ct) method (2^−ΔΔCt^) [19].

### 2.7. Plant Transformation and Generation of Transgenic Plants

The amplified CDS of *MsVDAC* was inserted into the BamHI/PstI sites of a pCAMBIA1300-35S-sGFP vector, yielding a pCAMBIA1300-35S::MsVDAC expression vector construct. Then, the recombinant pCAMBIA1300-35S::MsVDAC vector was introduced into *A. tumefaciens* strain EHA105, which was then transformed into wild-type (WT) tobacco by the agroinfiltration method. More than 30 independent lines were selected by hygromycin (Hyg) resistance. The 20 mg/L hygromycin was added to the seed germination medium for screening and the gene-specific primers of Hyg (forward primer, 5′-ACGGTGTCGTCCATCACAGTTTGCC-3′, reverse primer, 5′-GGAAGTGCTTGACATTGGGGAGTTT-3′) and *MsVDAC* (forward primer, 5′-ATGGCTAACGGTCCAGCAC-3′, reverse primer, 5′-TTAAGGCTTGAGAGAAAGAGCG-3′) to identify the positive transgenic plants via PCR before sampling the successfully grown plants. The homozygous transformants (T2) were further confirmed via qPCR using specific primers for *MsVDAC* (same as the qPCR primers of *MsVDAC* in Section 2.6), and the *N. tabacum Actin* gene (*NtActin*, Appendix A) was used as an internal reference. T2-generation homozygous lines, including line 1 (V-1), line 2 (V-2), line 7 (V-7), and WT lines were used for further stress tolerance assays and expression analysis.

### 2.8. Stress Tolerance Assays

Seeds of WT and transgenic tobacco lines were sown on 1/2-strength MS medium plates and incubated at 4 °C for 2 days before being placed in a growth chamber at 25 °C for 7 days. The geminated seedlings were subsequently transplanted into plastic pots filled with vermiculite in a greenhouse. Three-week-old plants were used for stress tolerance assays. For cold tolerance assays, the plants were placed in a growth chamber at 4 °C for 7 days. For drought tolerance assays, the plants were watered with nutrient solution containing 20% PEG6000 for 7 days. For salt tolerance assays, the plants were watered with nutrient solution that included 150 mM NaCl for 7 days. Then, the leaves of the treated plants were harvested immediately, frozen in liquid nitrogen and stored at −80 °C for further analysis.

### 2.9. Measurement of Physiological Changes

To measure the physiological changes in WT and transgenic tobacco, the leaves of each sample were ground in liquid nitrogen and measured in a spectrophotometer. The accumulation of malondialdehyde (MDA) was determined according to the methods of Li et al., which involved the use the modified thiobarbituric acid (TBA) method [20]. The superoxide anion radical (O_2_
^-^) content was determined as described by Elstner [21], and the activities of peroxidase (POD) and superoxide dismutase (SOD) were determined based on the changes in absorption values at 470 nm and 560 nm, respectively [22]. The content of reduced glutathione (GSH) was fluorometrically estimated [23]. The content of soluble sugar (SS) was determined using the methods of Dreywood [9], and the soluble protein (SP) content was determined by the Bradford method [24]. The contents of free proline were determined according to the ninhydrin method [25].

### 2.10. Expression Analysis of Stress-Response Genes

To further illustrate the role of *MsVDAC* in stress tolerance, the expression patterns of 10 stress-response genes were analysed using qPCR. These ten genes included calcineurin B-like protein 1 (*CBL1*), glyceraldehyde-3-phosphate dehydrogenase, and cytosolic protein (*GAPC*). Moreover, Bax inhibitor 1-like (*BI-1*), hexokinase-2-like (*Hxk3*), protein gamma response 1 (*GR1*), superoxide dismutase [Cu-Zn] 2 (*Cu/Zn-SOD*), superoxide dismutase [Mn] (*MnSOD*), nonspecific lipid-transfer protein 1 (*Ltp1*), dehydrin DHN1-like (*ERD10B*), and serine/threonine-protein kinase (*SnRK2*) have been demonstrated to play key roles in abiotic stress tolerance, and most of these proteins were associated with the changes in physiological indicators measured above. *NtActin* was used as an internal reference gene, and the primers used in this experiment are listed in Appendix A.

### 2.11. Statistical Analysis

All the experiments described in this research were repeated three times, with three biological replicates. Student’s *t*-test and one-way analysis of variance were applied to determine significant differences for statistical analysis with SPSS 22.0 statistical software.

## 3. Results

### 3.1. Isolation and Characteristics of MsVDAC

The VDAC domain of *A. thaliana* was used to blast the genome of “*M.sativa* L. cv. Zhongmu No.1” and 13 VDAC homologous proteins were found. Three proteins without Porin3 domain verified by domain analysis were removed. Finally, a total of 10 VDAC homologous proteins were obtained, which belong to three different subclasses: VDAC1, VDAC2 and VDAC4, respectively (Appendix A). These three subclasses all contain Porin3 domain, and the differences among them were the number of exons (Appendix A). Herein, we isolated the full-length cDNA of *MsVDAC* (GenBank accession number: MW556128), which was 840 bp (Appendix A) and encoded a protein of 278 amino acids. The molecular formula of *MsVDAC* was C_1341_H_2117_N_361_O_413_S_4,_ with a calculated molecular weight of 30.03 kDa and pI of 9.19. The extinction coefficient of the *MsVDAC* protein solution at 280 nm was 17,420, and the instability coefficient of the protein was 25.09, with an average hydrophilic coefficient of −0.262, which indicates that it is a stable and hydrophilic mitochondrial protein. Alignment of the amino acid sequence of *MsVDAC* and of the eight other homologues indicated that *MsVDAC* is a Porin_3 protein (Appendix A). The phylogenetic tree constructed by the neighbour-joining method indicated that *MsVDAC* resided on an adjoining branch together with *MtVDAC* and was most homologous with *MtVDAC* (Appendix A). The predicted three-dimensional structure of *MsVDAC* showed that the *MsVDAC* protein consisted of 64 α-helices (22.94%), 19 β-turns (6.81%), 76 extended strands (27.24%), and 120 random coils (43.015%) (Appendix A).

### 3.2. Subcellular Localization of MsVDAC

To determine the subcellular localization of *MsVDAC*, the MsVDAC-GFP fusion protein was transiently coexpressed in *N. benthamiana* leaves, which were then observed via laser scanning confocal microscopy. As shown in Figure 1, MsVDAC-GFP fluorescence was distributed within the cytoplasm. In combination with the predicted subcellular localization of *MsVDAC* from CELLO 2.5 (Appendix A), our results indicated that the *MsVDAC* protein was localized mainly in the mitochondria of plant cells.

### 3.3. Expression Pattern of MsVDAC

The expression patterns of *MsVDAC* in different alfalfa tissues and in response to abiotic stress were examined via qPCR. The results showed that the expression level of *MsVDAC* in the alfalfa roots was higher than that in the leaves and inflorescences (Figure 2A). A significantly high level of *MsVDAC* expression in the roots was observed at 3 h (2.06-fold) under cold stress (Figure 2B). The increased expression of *MsVDAC* in the roots decreased significantly after 24 h (0.39-fold) under drought stress. In response to salt stress, the *MsVDAC* expression level in the roots peaked at 6 h (1.36-fold) and arrived at a nadir at 48 h (0.72-fold). In the leaves, the expression level of *MsVDAC* was significantly downregulated at 24 h (0.25-fold) under cold stress and tended to first decline (6 h), then increase (12 h) and then decline again (24 h) under drought stress. In response to salt stress, the expression level of *MsVDAC* in the leaves significantly increased, peaking at 6 h (27.00-fold), and subsequent gene expression slowly decreased from 12 to 24 h (Figure 2C). Taken together, these results suggested that the expression of *MsVDAC* is tissue specific in alfalfa and that the *MsVDAC* gene might be involved in the self-protection mechanism of alfalfa in response to adverse environmental conditions.

### 3.4. Overexpression of MsVDAC Improves the Cold, Drought and Salt Tolerance of Transgenic Tobacco

To investigate the role of *MsVDAC* in the plant response to abiotic stresses, the *MsVDAC* gene was overexpressed in tobacco via the recombinant vector pCAMBIA1300-35S::MsVDAC (Figure 3A). Transgenic tobacco plants (T2 generation) were generated and confirmed via qPCR analysis. The expression levels of *MsVDAC* in V-1, V-2, and V-7 were significantly higher than those in the WT (Figure 3B). In particular, the expression level of *MsVDAC* in V-2 was 1522.31 times higher than that in the WT. As shown in Figure 3C, compared with the WT, the transgenic tobacco grew better and was more tolerant to cold, drought and salt stresses (Figure 3D–F).

### 3.5. Physiological Changes in Tobacco Overexpressing MsVDAC under Cold, Drought and Salt Stress

To further understand the function of *MsVDAC* in abiotic stress tolerance, we investigated the physiological changes in our transgenic tobacco plants and WT plants. Cell membrane damage was determined by measuring the MDA content. As shown in Figure 4A, the MDA content in transgenic tobacco was significantly higher than that in the WT plants under cold and drought stress. However, the MDA content in V-1 and V-2 was lower than that in the WT plants under salt stress. The levels of oxidative damage and ROS scavenging in the WT plants and transgenic lines were evaluated by measuring the contents of O_2_
^-^, the activities of SOD and POD, and the content of GSH. The O_2_
^-^ content in the transgenic lines was significantly higher than that in the WT plants under cold stress; however, the content did not fluctuate much under drought stress and increased sharply in the V-2 plants under salt stress (Figure 4B). Although we did not observe significant changes in O_2_
^-^ content in the V-7 plants under abiotic stresses, the activities of SOD and POD in the V-1 plants were higher than those in the WT plants under salt stresses (Figure 4C,D). The activities of SOD and POD in the V-2 plants were significantly lower than those in the WT plants under cold stress but higher than those in the WT plants under salt stress. The content of GSH increased significantly in all three transgenic lines under cold stress (Figure 4E). However, only the GSH content in V-7 under drought stress greatly increased. The contents of SS, SP and proline were measured to evaluate the osmotic adjustment in the WT plants and transgenic tobacco. The SS, SP and proline contents in the transgenic lines obviously increased under cold and drought stress (Figure 4F–H). We observed notable changes in SS and proline contents only in the V-2 plants under salt stress, and no significant changes in SP content were observed in response to salt stress. The highest contents of SS, SP and proline were observed in the V-2 plants, of which the contents of SS and proline under drought stress and the content of SP under cold stress were 1567.7302 mmol·g^−1^ DW, 131.8593 µmol·g^−1^ DW and 89.2847 mg·g^−1^ DW, respectively.

### 3.6. Expression Analysis of Stress-Response Genes in Transgenic Tobacco Plants Overexpressing MsVDAC under Cold and Drought Stress

According to the physiological changes in transgenic tobacco plants overexpressing *MsVDAC* under cold, drought and salt stress, we found that the expression of *MsVDAC* was more obvious under low temperature and drought stress. Thus, to further elucidate the possible regulatory mechanism of *MsVDAC*, we measured the expression of 10 stress-response genes (*Hxk3*, *CBL1*, *GR1*, *GAPC*, *Cu/ZnSOD*, *MnSOD*, *BI-1*, *Ltp1*, *ERD10B* and *SnRK2*) via qPCR assays in the transgenic lines and WT plants under cold and drought stress. As shown in Figure 5, the expression levels of *Hxk3* and *Ltp1* increased in MsVDAC-overexpressing tobacco under cold and drought stress, but they were significantly lower than those in the WT plants under the control conditions. The expression level of *Hxk3* increased significantly in the V-2 and V-7 plants under cold stress and increased significantly in the V-1 and V-2 plants under drought stress. The expression levels of Ltp1 increased markedly in all three transgenic lines under cold stress but showed a significant increase only in the V-7 plants under drought stress. The expression levels of *CBL1*, *GR1*, *GAPC*, and *BI-1* in the transgenic lines were lower than those in the WT plants under both normal conditions and drought stress conditions but were significantly higher in the V-1, V-2, V-7, and V-2 plants under cold stress than in the WT plants. The expression levels of *Cu/ZnSOD* in the transgenic lines were higher than those in the WT plants under both the control and cold stress conditions, but they were lower in all three transgenic lines than in the WT plants under drought stress conditions. The expression levels of *MnSOD* in the transgenic lines were also lower than those in the WT plants, but no obvious difference was observed. The expression level of *MnSOD* in the V-2 plants significantly increased under cold stress. However, we did not observe significant changes in *SnRK2* expression in the transgenic lines compared with the WT plants under cold or drought stress. The expression levels of *ERD10B* in the V-2 and V-7 plants were significantly higher than those in the WT plants under cold and drought stress.

## 4. Discussion

VDACs are the most abundant proteins in the outer mitochondrial membrane and constitute the major transport pathway between mitochondria and the cytoplasm [26]. In plants, VDACs not only play a role in metabolite transport but are also involved in programmed cell death triggered in response to abiotic and biotic stresses [27]. Although many plant *VDAC* genes that participate in temperature, oxidative stress and salt stress responses have been characterized [9,28], there have been no studies on these genes in alfalfa. In this study, a *VDAC* gene from alfalfa named *MsVDAC* was cloned and functionally characterized. *MsVDAC* encodes a 278-amino-acid protein containing a conserved porin-3 domain (Appendix A). It is similar to the *VDAC* of *M. truncatula* [29] and is located in the mitochondria, as is the case for *VDACs* in Arabidopsis, tobacco and pea [10,12] (Figure 1). The expression level of *MsVDAC* was higher in the leaves and roots and lower in the inflorescences (Figure 2A), which was consistent with the expression of *BrVDAC* in *Brassica rapa* [11]. The expression level of *MsVDAC* showed significant differences under cold, drought, and salt stress (Figure 2B,C), indicating that *MsVDAC* is involved in the response of alfalfa to abiotic stress.

Physiological changes in the transgenic lines were measured to further characterize the function of *MsVDAC* under abiotic stress. The MDA content was evaluated because it is an important indicator of membrane injury under abiotic stress [30]. Active oxygen, as a by-product of aerobic metabolism, can have a toxic effect on plants [31]. The enzymatic antioxidant system is a protective mechanism that can eliminate or quench ROS and improve the ability of plants to tolerate abiotic stress [32]. Our data showed that, compared with the WT plants, the transgenic plants had higher MDA and O_2_
^-^ contents but lower SOD and POD activities under cold and drought stress. Overexpression of *TaSAUR78* was shown to improve plant tolerance to abiotic stresses by regulating *TaVDAC1*, and overexpression of *TaVDAC1* increased the accumulation of ROS in transgenic *Arabidopsis* [33]. The results of the present study confirmed that *MsVDAC* plays a negative regulatory role in the resistance of alfalfa to low temperature and drought. As osmotic regulators, free proline, SS and SP can increase the osmotic potential of cells, protect the stability of biofilms, reduce oxidative damage, and enhance plant tolerance to environmental conditions [34]. Here, we observed that the contents of proline, SS, and SP in the overexpression lines were significantly higher than those in the WT plants, indicating that *MsVDAC* can participate in the response mechanism of plants to low temperature and drought.

The expression of stress-responsive genes can be induced in response to abiotic stress. Hexokinase, a functional regulator of *VDAC*, has been proven to play a role in pathogen induction and H_2_O_2_-induced cell death in plants [35]. Metabolomics analysis of *A. thaliana* ectopically expressing the *Hxk3* gene showed that phosphorylated sugar (G6P), starch and some metabolites related to the tricarboxylic acid (TCA) cycle are affected [36]. *LTP1* plays a role in fatty acid binding and phospholipid and lipid transfer between membranes in vitro [37] and can be strongly induced in *Lotus japonicus* in response to drought stress [38] and in wheat under cold stress [39]. The expression level of *ERD10B* responds to both ionic and osmotic stress [40]. In our study, the expression of *Hxk3*, *LTP1* and *ERD10B* was significantly upregulated under low temperature and drought stress (Figure 5). Under the same conditions, the *MsVDAC* transgenic plants also showed increased levels of dehydrins, which act as structural stabilizers with chaperone-like properties and protect various nuclear and cytoplasmic macromolecules from coagulation during dehydration [41]. These results indicated that the overexpression of *MsVDAC* can increase the expression of *ERD10B*, *LTP1* and *Hxk3* under low temperature and drought stress, thereby regulating the ROS clearance mechanism and lipid conversion and the contents of glucose, sucrose, and starch, thereby affecting photosynthesis and the TCA cycle in alfalfa.

*Cu/ZnSOD* and *MnSOD* are two subtypes of SOD and are considered to be the first and most important lines of defence of antioxidant enzyme systems in most organisms [42]. The senescence process of peach was shown to be accompanied by an increase in *VDAC* content, loss of catalytic activity of *MnSOD* and promotion of the release of superoxide anions [43]. *GR1*, an antioxidant enzyme-encoding gene, participates in the ascorbate-glutathione (AsA-GSH) cycle and can effectively maintain appropriate levels of reduced GSH to relieve oxidative stress [44,45]. In this study, the O_2_
^-^ content in the transgenic plants increased significantly under cold stress, and the activities of SOD and POD decreased significantly in the V-2 lines, which may be related to the significant upregulation of *Cu/ZnSOD* and *MnSOD*. The content of GSH in V-2 was also significantly upregulated with increasing O_2_
^-^ content, while the expression of *GR1* remained stable. However, the expression of *Cu/ZnSOD* in the transgenic lines was significantly downregulated under drought stress, and there was no significant difference in the content of O_2_
^-^, in the activity of SOD, or in the expression of *MnSOD*. Therefore, the overexpression of *MsVDAC* needs to be higher for this gene to participate in the regulation of the cold tolerance of plants by regulating the antioxidant enzyme system, and the regulation of drought resistance by *MsVDAC* may not occur mainly through the regulation of the antioxidant enzyme system.

*CBL1* is a positive regulator of the drought response and a negative regulator of the cold response in plants [46]. *VDAC1* can interact with *CBL1*, and both regulate cold stress responses [47]. In this study, the expression of *CBL1* in the transgenic tobacco was significantly higher than that in the WT plants under cold stress but significantly lower than that in the WT plants under drought stress, indicating that the overexpression of *MsVDAC* can revert the regulation of the expression of *CBL1* under cold stress and adversely regulate the expression of *CBL1* under drought stress. *GAPC1* and *GAPC2* can also interact with *VDAC3*, and the glycolytic enzyme GAPDH seems to bind to the outer mitochondrial membrane in a redox-dependent manner [48]. GAPCs are believed to be key enzymes involved in the glycolysis and gluconeogenesis metabolic pathways [49]. Overexpression of *TaGAPC1* and *TaGAPC2* can improve the drought tolerance of *Arabidopsis* [50,51], and the constitutive expression of *StGAPC1*, *StGAPC2*, and *StGAPC3* in potato tubers under cold stress could promote the accumulation of reducing sugars through the sucrose pathway by altering the accumulation of tuber metabolites [52]. Our results showed that the overexpression of *MsVDAC* could increase the expression of *GAPC* under cold stress but decrease the expression of *GAPC* under drought stress. Herein, the upregulation of *GAPC* under cold stress may increase ATP and abscisic acid (ABA) contents by regulating the glycolysis pathway. However, the regulatory effect may be the opposite under drought stress. The expression of *BI-1*, a conserved protein that plays a cytoprotective role in the regulation of cellular processes [53], also showed similar changes in transgenic tobacco in response to both cold and drought conditions. Therefore, the regulatory mechanism of *MsVDAC* may differ under cold and drought conditions. *SnRK2* is a member of the ABA signalling pathway and is involved in the regulation of ABA in response to plant stress [54], and the overexpression of *TaSnRK2.3* can enhance tolerance to drought, salt and freezing stress [55]. However, we found no significant changes in the expression of *SnRK2* under cold and drought conditions, which was consistent with the finding that exogenous ABA had no effect on the expression of *PgVDAC* in Forever Blue pennisetum [28]. Here, the responses of *MsVDAC* under cold and drought stress may not occur through the ABA regulatory pathway.

Therefore, we hypothesized that the role of *MsVDAC* in the plant response to cold and drought stress may involve a complex regulatory network that includes changes in both physiology and gene expression (Figure 6). The overexpression of *MsVDAC* enhanced the cold tolerance of plants; led to enhanced ROS scavenging via increased contents of GSH and expression of *GAPC*, *BI-1*, *Cu/ZnSOD* and *MnSOD*; and maintained osmotic homeostasis by increased contents of MDA, SS and SP and expression of *Hxk3*, *ERD10B* and *LTP1*. However, the enhancement of *MsVDAC* in transgenic plants may occur only through osmotic homeostasis, since the expression of *CBL1*, *BI-1* and *GAPC* was downregulated in MsVDAC-overexpressing tobacco plants.

## 5. Conclusions

In summary, we cloned *MsVDAC*, a novel VDAC-encoding gene, from alfalfa and identified its positive role in the improvement of cold and drought tolerance. *MsVDAC* localized to the mitochondria, and its expression was highest in alfalfa roots and was induced in response to cold, drought and salt treatment. Overexpression of *MsVDAC* in tobacco significantly increased the contents of MDA, GSH, SS, SP and proline under cold and drought stress. However, the activities of SOD and POD decreased in transgenic tobacco plants under cold stress, while the content of O_2_
^-^ increased. Stress-responsive genes including *LTP1*, *ERD10B* and *Hxk3* were upregulated in transgenic plants under cold and drought stress. However, *GAPC*, *CBL1*, *BI-1*, *Cu/ZnSOD* and *MnSOD* were upregulated only in transgenic tobacco plants under cold stress, and *GAPC*, *CBL1*, and *BI-1* were downregulated under drought stress. These results suggest that *MsVDAC* provides cold tolerance by regulating ROS scavenging, osmotic homeostasis and the expression of stress-responsive genes in plants, but the improved drought tolerance via *MsVDAC* overexpression may be mainly due to osmotic homeostasis and the expression of stress-responsive genes in plants. Our study provides new insight into how VDACs improve cold and drought stress in plants and how *MsVDAC* can be targeted in future breeding programmes to improve cold and drought tolerance in alfalfa.

## Figures and Tables

**Figure 1 genes-12-01706-f001:**
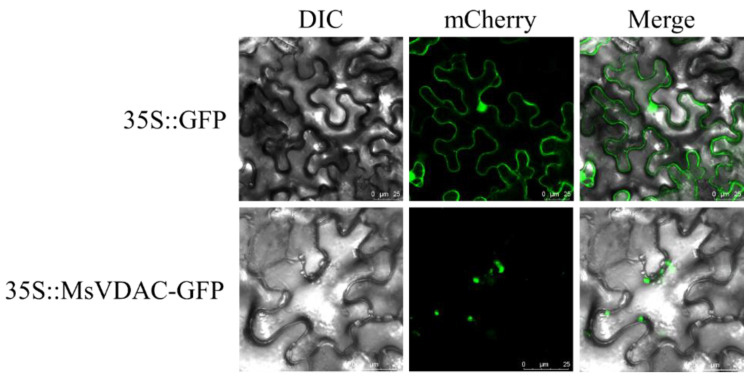
Subcellular localization of *MsVDAC*. The recombinant vector MsVDAC-GFP and 35S:GFP control vector were transferred into epidermal cells of *N. benthamiana* leaves. Scale bar, 25 μm.

**Figure 2 genes-12-01706-f002:**
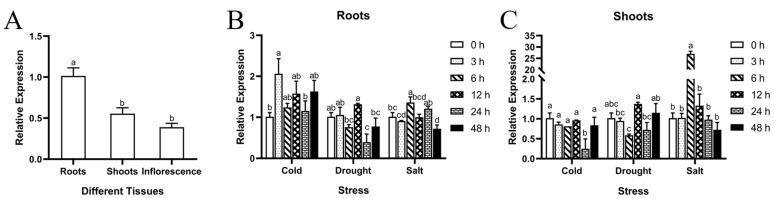
Expression analysis of *MsVDAC* in alfalfa: (**A**) Tissue-specific expression of *MsVDAC* in alfalfa roots, leaves and inflorescences; (**B**) Expression pattern of *MsVDAC* in alfalfa roots under cold, drought and salt stress; (**C**) Expression patterns of *MsVDAC* in alfalfa leaves under cold, drought and salt stress. The error bars represent the means ± SDs of three independent biological replicates. The different letters represent significant differences (α = 0.05).

**Figure 3 genes-12-01706-f003:**
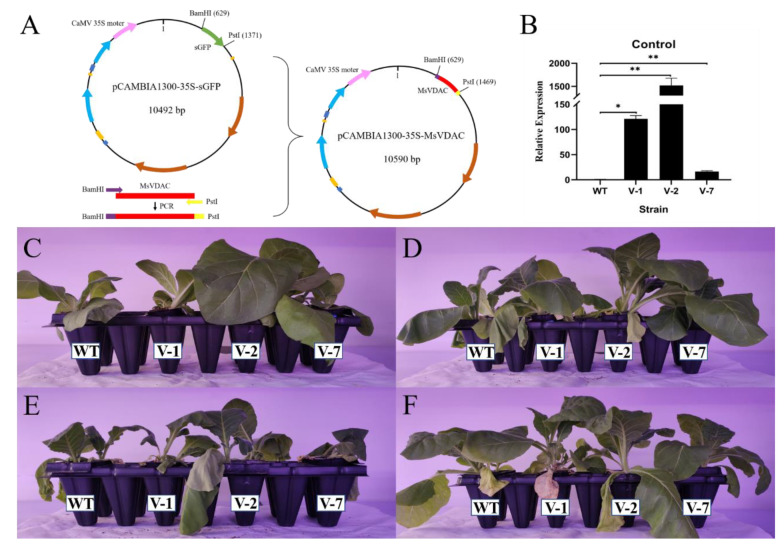
(**A**) Construction of the pCAMBIA1300-35S::MsVDAC recombinant vector; (**B**) Expression level of *MsVDAC* in transgenic tobacco (T2 generation) and WT plants according to qPCR analysis; (**C**–**F**) Phenotypes of transgenic tobacco and WT plants under normal conditions and cold, drought, and salt stress, respectively. WT: wild type. V-1: line 1. V-2: line 2. V-7: line 7. *, *p* < 0.05. **, *p* < 0.01.

**Figure 4 genes-12-01706-f004:**
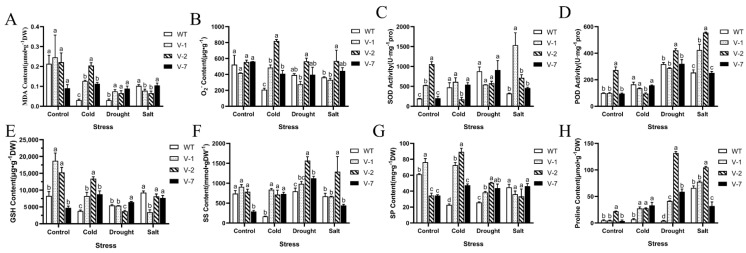
Physiological analysis of the cold, drought and salt stress tolerance of transgenic tobacco plants overexpressing *MsVDAC*: (**A**) MDA content; (**B**) Superoxide anion radical (O_2_ ^-^) content; (**C**) SOD activity; (**D**) POD activity; (**E**) Reduced GSH content; (**F**) SS content; (**G**) SP content; (**H**) Free proline content. WT: wild type. V-1: line 1. V-2: line 2. V-7: line 7. The error bars represent the means ± SDs of three independent biological replicates. The different letters represent significant differences (α = 0.05).

**Figure 5 genes-12-01706-f005:**
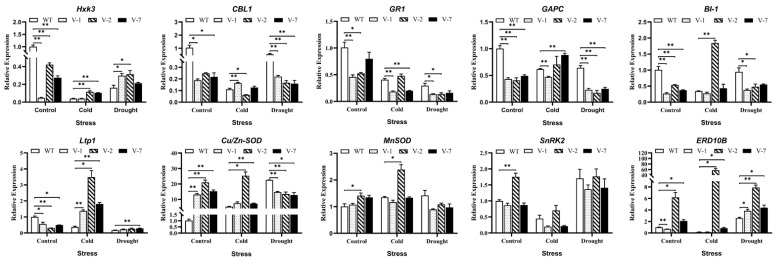
Expression of 10 stress-response genes in transgenic tobacco plants overexpressing *MsVDAC* under cold and drought stress. WT: wild type. V-1: line 1. V-2: line 2. V-7: line 7. The error bars represent the means ± SDs of three independent biological replicates. *, *p* < 0.05. **, *p* < 0.01.

**Figure 6 genes-12-01706-f006:**
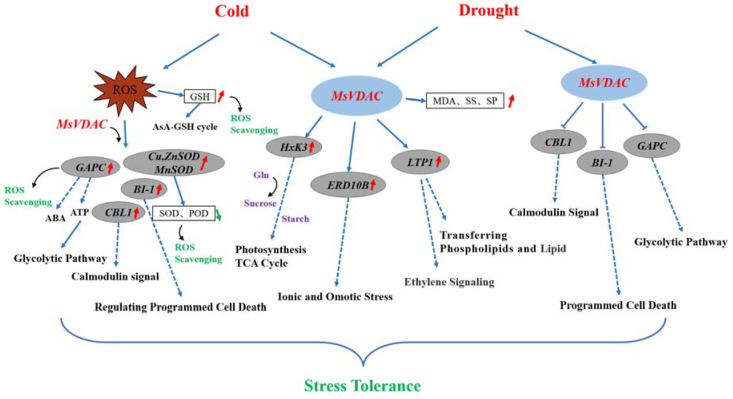
Possible molecular mechanisms through which *MsVDAC* functions in response to cold and drought stress. The red arrows indicate upregulation of stress-response genes or increases in physiological indicators. The possible regulatory mechanisms are shown with dotted lines.

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
