# Peer review of "Overexpression of a Voltage-Dependent Anion-Selective Channel (VDAC) Protein-Encoding Gene, MsVDAC, from Medicago sativa Confers Cold and Drought Tolerance to Transgenic Tobacco"

_genes, 2021, doi:10.3390/genes12111706_

Round 1

Reviewer 1 Report

Dear authors,

I have read the manuscript entitled “Overexpression of a voltage-dependent anion-selective channel (VDAC) protein-encoding gene, MsVDAC, from Medicago sativa confers cold and drought tolerance to transgenic tobacco” and found it interesting. You inserted a construct for an enhanced expression levels of MsVDAC in tobacco, subjected the plants to different environmental cues, and observed their response. I believe there are important findings to be shared for researchers working on alfalfa abiotic stress tolerance. I have the following comments and questions to ask.

Comments:

There are inconsistent uses of abbreviations. For example, VADC vs VDAc (Line 51, 76, 77, 416 vs 12, 15, 44, and so forth).

It is mentioned on line 126 and 128 that the enzyme restriction sites are underlined, but I couldn’t see underline. Go through the whole manuscript and edit the inconsistencies, typos, improper expressions.

Questions:

Do you expect a compromised stress tolerance with a silenced/reduced expression levels of MsVDAC? have you assessed but not reported here the reduced levels of MsVDAC on the plant performance upon the environmental cues?

Do you expect the same response if the expression levels of MsVDAC is altered in alfalfa compared to the response in tobacco?

How many predicted homologs of MsVDACs are in alfafa?

With the recently available alfalfa genome sequence can you able to characterize the MsVDAC homologs, at least in silko?

Do you expect an additive or redundant functions among the homologs?

Reviewer 2 Report

Authors present important and novel data related to functions of MsVDAC protein in transgenic tobacco in response to cold, drought and salt treatment. Some improvements of manuscript are proposed as follows:

Section 2.4

The underlined sequences recognized by both restriction enzymes are not visible.

Escherichia coli, Agrobacterium tumefaciens- should be in italics.

Section 2.5

How the RNA was isolated?

Assessment of RNA purity, concentration and integrity

Details of DNase treatment to remove putative remnants of genomic DNA

Reverse transcription reaction; details of reaction-temperature and time, volume and amount of used RNA

qPCR target- length of PCR product (control-actin and tested gene), target gene symbol and accession number.

qPCR protocol: conditions of PCR reaction, volume of reaction, concentration of magnesium ions, dNTPs, DNA polymerase type and concentration.

Reason to choose particular internal (Msactin) standard- citation of previous research or analysis using bestKeeper tool (Pfaffl et a. 2004)

Software and equipment used to perform and analyse RT-PCR results.

Section 2.6

Concentration of hygromycin applied to select transformants.

  1. tabacum, A. tumefaciens- should be in italics.

How the homozygosity of T2 plants was assured?

Section 3.1

Is the 3D structure of MsVDAC related to structure of homologous proteins?

Section 3.2

Maybe Authors could experimentally confirm presence of MsVDAC in mitochondria by using uncoupling agents as FCCP or 2,4-DNP to inhibit mitochondria function, destroy it and change distribution of MsVDAC?

Alternatively Authors may use fluorescence stain that are specific for mitochondria as for example  rhodamine 123, tetramethylrosamine, mitotracker green, red or orange, to show colocalization of MsVDAC and mitochondria-specific stain.

Author Response

Please see the attachment,thank you.

Round 2

Reviewer 2 Report

In my opinion Authors properly answered to presented suggestions and accordingly corrected manuscript. Some doubts related to experimental localization of MsVDAC in mitochondria remain, however Authors explained their approach by a similar methodology applied in earlier studies.